# A Novel Role of the Two-Component System Response Regulator UvrY in Enterohemorrhagic *Escherichia coli* O157:H7 Pathogenicity Regulation

**DOI:** 10.3390/ijms24032297

**Published:** 2023-01-24

**Authors:** Pan Wu, Qian Wang, Qian Yang, Xiaohui Feng, Xingmei Liu, Hongmin Sun, Jun Yan, Chenbo Kang, Bin Liu, Yutao Liu, Bin Yang

**Affiliations:** 1TEDA Institute of Biological Sciences and Biotechnology, Nankai University, Tianjin 300457, China; 2Key Laboratory of Molecular Microbiology and Technology, Ministry of Education, Nankai University, Tianjin 300457, China; 3Nankai International Advanced Research Institute, Nankai University, Shenzhen 518000, China

**Keywords:** enterohemorrhagic *Escherichia coli* (EHEC) O157:H7, two-component systems (TCSs), UvrY, LEE genes, pathogenicity

## Abstract

Enterohemorrhagic *Escherichia coli* (EHEC) O157:H7 is an important human pathogen causing severe diseases, such as hemorrhagic colitis and lethal hemolytic uremic syndrome. The signal-sensing capability of EHEC O157:H7 at specific host colonization sites via different two-component systems (TCSs) is closely related to its pathogenicity during infection. However, the types of systems involved and the regulatory mechanisms are not fully understood. Here, we investigated the function of the TCS BarA/UvrY regulator UvrY in the pathogenicity regulation of EHEC O157:H7. Our results showed that UvrY acts as a positive regulator of EHEC O157:H7 for cellular adherence and mouse colonization through the transcriptional activation of the locus for enterocyte effacement (LEE) pathogenic genes. Furthermore, this regulation is mediated by the LEE island master regulator, Ler. Our results highlight the significance of UvrY in EHEC O157:H7 pathogenicity and underline the unknown importance of BarA/UvrY in colonization establishment and intestinal adaptability during infection.

## 1. Introduction

Enterohemorrhagic *Escherichia coli* (EHEC) O157:H7 is one of the most common EHEC pathogenic bacteria. Normally, 10–100 colony-forming units of EHEC O157:H7 are sufficient to cause colon infection [1]. The common clinical symptoms of pathogenic EHEC O157:H7 infection include diarrhea, hemorrhagic colitis (HC), and systemic hemolytic uremic syndrome (HUS) [2]. The pathogenic regulatory mechanism of EHEC O157:H7 has been intensively investigated before, and the type III secretion system (T3SS) and Shiga toxin are crucial virulence factors [3,4]. T3SS is controlled by several effectors and regulators, which are encoded by 41 genes located at the LEE pathogenicity islands, operons LEE1–LEE5 [5,6]. LEE operons are highly effective adherence operons that function for EHEC O157:H7 initiating and establishing intimate adherence in host cells [7]. The first gene of the LEE1 operon (*ler)* encodes the LEE master regulator Ler (LEE-encoded regulator) [5,6]. T3SS also contributes to attaching and effacing (A/E) lesions, which are characterized by the loss of microvilli from the intestinal brush border and the formation of an intimate attachment of the bacterium to the host cell [5]. To prevent energy burden in the efficient adaptation or pathogenicity execution in fluctuating environmental conditions, EHEC O157:H7 has evolved elaborate paths in the transcriptional regulation of the LEE island genes [8,9] in a Ler-dependent [5,7,10] or independent manner [11]. Considering that LEE regulation is often orchestrated by upstream factors, the demonstration of the regulation pathway would be pivotal for pathogenic EHEC infection control in the clinic.

Two-component systems (TCSs), consisting of histidine sensor kinase and a response regulator [12], are pivotal for signal transduction [13]. Typically, signal sensing through the N-terminus portion of the membrane-anchored sensor protein results in the phosphorylation of specific histidine residues, followed by a process in which a sensor transduces the phosphate group to an aspartate residue on the response regulator protein, which enables the regulator to affect target gene transcription activity [13]. The function of TCSs in virulence regulation has been described in many pathogenic bacteria [14], such as the identified GacA/GacS system in *Pseudomonas aeruginosa* [15], BprS/BprR in *Burkholderia pseudomallei* [16], and CarS/CarR TCSs in *Vibrio cholerae* [17]. In EHEC O157:H7, several TCSs’ sensing intestinal signals have been characterized for colonization regulation and pathogenicity, among which PhoQ/PhoP, BaeS/BaeR, QseB/QseC, and RstA/RstB have pathogenicity elevation roles, whereas CpxA/CpxR and FusK/FusR play repressive functions in virulence, respectively [8,18,19,20,21,22]. However, how the versatile environmental signals are transduced and regulated through different TCSs and, hence, modulate the virulence capability in EHEC O157:H7 is still not fully uncovered.

The BarA/UvrY TCS comprises the transmembrane sensor kinase protein BarA and the cytoplasmic response regulator protein UvrY. The functions of the BarA/UvrY orthologs have been demonstrated in several other *E. coli* lineage strains [23]. The *uvrY* mutation leads to a reduction in biofilm formation and an elevation in bacterial motility in the K12 strain [24,25]. UvrY function in switching gluconeogenesis and glycolysis through the small RNAs *csrB* and *csrC* has been described in K12 [26,27]. Moreover, the contribution of UvrY to pathogenicity has been reported. For example, UvrY promotes the expression of type I fimbriae, which, in turn, increases intestinal colonization in the adherent-invasive *E. coli* (AIEC) strain LF82 [28]. In the uropathogenic *E. coli* (UPEC) strain CFT073, the mutation of *uvrY* reduces the production of lipopolysaccharide (LPS) and leads to a repression effect on gene coding for the host uroepithelial cellular cytokines TNF-α and IL-6 [29], indicating the positive role of UvrY in UPEC virulence. In addition, the role of the BarA/UvrY ortholog in pathogenicity has also been described in other species, such as ExpS/ExpA in *Erwinia* spp., GacS/GacA in *Pseudomonas* spp., and BarA/SirA in *Salmonella enterica* [30,31]. However, the effect of UvrY on EHEC O157:H7 pathogenicity remains largely unknown.

In this study, we revealed that UvrY is crucial for the pathogenicity of EHEC O157:H7. Both ex vivo and in vivo experiments showed that UvrY plays a positive role in bacterial adherence to the infected host. The adhesion promotion and A/E lesion formation capability of UvrY was further demonstrated by the positive regulation of LEE gene expression through Ler. The whole genome transcriptome in our study also suggests a global function of UvrY in EHEC O157:H7 for virulence modulation and environmental adaptation. Overall, this study characterizes the novel role of the response regulator UvrY in EHEC O157:H7 pathogenicity and demonstrates its importance in virulence and host colonization through an unclear upstream sensing signal or regulation mechanism that needs to be further understood.

## 2. Results

### 2.1. UvrY Has a Positive Effect on the Adherence of EHEC O157:H7 to Its Infected Host

To investigate whether UvrY affects EHEC O157:H7 pathogenicity, an ex vivo cellular adherence assay was performed in HeLa cells in the wild type (WT) and Δ*uvrY* strains. The Δ*uvrY* strain displayed a 9.78-fold reduction in its adherence capability after co-culturing with HeLa cells for 3 h compared with the WT, and the adhesion phenotype could be restored upon complementation with the *uvrY* gene (Figure 1A), indicating the importance of UvrY in bacterial adherence to the host. Furthermore, to rule out the possibility that the adherence reduction effect caused by *uvrY* deletion was due to its decreased growth rate, the growth curves of the WT, Δ*uvrY*, and Δ*uvrY*+ strains were measured. No significant difference was detected for the WT, Δ*uvrY*, and Δ*uvrY*+ strains in 24 h growth monitoring in a virulence-inducing medium, Dulbecco’s modified eagle medium (DMEM) [32], or LB broth in vitro (Figure 1B and Appendix A), indicating that the reduction in the adherence ability of the Δ*uvrY* strain was not attributed to growth defects but due to an unclear mechanism that needs to be further clarified.

A/E lesion formation is the most important pathogenic feature after the EHEC adherence to host cells, during which the pedestal-like structures are established on the luminal surface of intestinal epithelial cells [7]. To test whether *uvrY* deletion affects pedestal formation, an in vitro fluorescent actin staining (FAS) assay was performed. The result showed that the proportion of infected cells significantly decreased upon *uvrY* deletion relative to the WT strains, which was approximately 50% (Figure 1C,D), revealing the positive role of UvrY in EHEC O157:H7 pathogenicity. Consistently, the Δ*uvrY* strain infection resulted in a significantly decreased pedestal formation number (~30%) in HeLa cells compared to the WT (Figure 1E). Both the infection percentage and pedestal formation levels in infected cells could extend to the level observed in the WT upon *uvrY* gene complementation (Figure 1C–E), demonstrating that UvrY is critical for EHEC O157:H7 in the cellular adherence capacity.

Furthermore, to understand whether the UvrY effect on cellular adherence can imitate the in vivo host infection process, a mouse intestinal colonization assay following EHEC O157:H7 inoculation was performed. The colony number of EHEC O157:H7 recovered from the mouse colon 6 h after inoculation was remarkably decreased (22.08-fold) upon *uvrY* deletion (Figure 1F). Complementing the *uvrY* gene could endow the Δ*uvrY* strain with regaining WT colonization levels (Figure 1F), reinforcing the crucial function of UvrY in host adherence. These results indicated that UvrY promotes the EHEC O157:H7 colonization level and adherence capabilities inside the host.

### 2.2. UvrY Promotes Host Colonization by LEE Gene Expression Induction in EHEC O157:H7

Considering the pivotal importance of LEE islands in EHEC O157:H7 intimate adherence [33,34], whether the effect of UvrY on host adherence is through regulating the LEE islands’ genes was measured. RNA from the WT, Δ*uvrY*, and Δ*uvrY*+ strains cultured in DMEM to log phase was extracted and subjected to qRT-PCR analysis. Interestingly, the expression of seven representative LEE genes measured (i.e., *ler*, *tir*, *eae*, *espB*, *escC*, *escN*, and *escT*) were downregulated in the Δ*uvrY* strain compared to the WT, but the expression level of the LEE genes between the WT and Δ*uvrY*+ strains are comparable (Figure 2A), indicating the importance of UvrY in LEE gene expression induction.

To investigate whether UvrY also induces LEE gene expression upon EHEC O157:H7 infection in the host, RNA was isolated from mouse colonic tissue after inoculation with the WT, Δ*uvrY*, and Δ*uvrY*+ strains at 6 h post-infection. Consistently, a significantly reduced level (5.3–14.9 fold) was detected in the Δ*uvrY* strains compared with the WT residing in colonic tissue, as measured by qRT-PCR (Figure 2B). However, when a functional *uvrY* gene was complemented to the *uvrY* mutant, this exhibited the same expression level as the WT (Figure 2B). These results demonstrated that UvrY promotes LEE gene expression both in vitro and in vivo. Furthermore, Western blotting analysis was performed for the protein level detection of the outer membrane (OM) adhesin intimin (encoded by the LEE gene *eae*) and its translocated receptor (encoded by the LEE gene *tir*), which are necessary for the intimate adherence of bacteria to host epithelial cells [34]. A consistent decrease in intimin and Tir (2.1–2.9 fold) was observed (Figure 2C,D), as the detected gene expression changes in the Δ*uvrY* strains (Figure 2A). Meanwhile, the Δ*uvrY*+ strains abolished the defect of Δ*uvrY* in the protein production of intimin and Tir (Figure 2C,D), confirming that UvrY activates LEE gene expression at both the RNA and protein levels. Therefore, our results showed that UvrY promotes host colonization by activating LEE-encoded T3SS.

Considering that UvrY is the cognate response regulator of the bacterial two-component system BarA/UvrY [23], an adherence assay and qRT-PCR analysis were performed to investigate whether the UvrY-mediated activation of EHEC O157:H7 pathogenicity occurs in a BarA-dependent manner. The results showed that the Δ*barA* strain displayed a significant reduction in adherence capacity and LEE gene expression compared with the WT (Appendix A), suggesting that the UvrY-mediated activation of EHEC O157:H7 pathogenicity occurs in a BarA-dependent manner.

### 2.3. UvrY Activates LEE Gene Expression via Ler

Considering that Ler, encoded by the LEE1 operon gene *ler*, is the master regulator and activator of the LEE operons [7], we contemplated whether the function of UvrY activation in LEE gene expression is Ler-dependent. To understand this, the Δ*ler*, Δ*ler*Δ*uvrY*, and Δ*ler*Δ*uvrY*+ strains were constructed, and an adherence assay was performed for these strains. In agreement with a previous report [35], the Δ*ler* strain exhibited a remarkably decreased adherence capacity compared to the WT strain in HeLa cells. However, there was no significant difference in the adherence capacity of the Δ*ler* and Δ*ler*Δ*uvrY* strains and the *uvrY* complementation strain Δ*ler*Δ*uvrY*+ (Figure 3A), indicating that UvrY did not provide an additional adherence capacity.

Consistent with the adherence results, reduced LEE gene expression was also detected through qRT-PCR analysis (Figure 3B), as well as the intimin and Tir protein level reduction (Figure 3C,D) in the Δ*ler*, Δ*ler*Δ*uvrY*, and Δ*ler*Δ*uvrY*+ strains compared with the WT. The results revealed that the cellular adherence capacity effect of UvrY is mediated by Ler. Furthermore, the results were reinforced in vivo through colon colonization experiments in mice. After orally inoculation with the EHEC O157:H7 strains for 6 h, the different *ler* mutant strains showed an approximately 42-fold lower bacterial load level than the WT-inoculated mice, regardless of whether the *uvrY* was depleted or complemented (Figure 3E). Collectively, these results demonstrated that UvrY activates LEE expression in a Ler-dependent manner.

### 2.4. UvrY Activates Ler Gene Expression Directly

Given that the regulation of LEE genes by UvrY is mediated by Ler, and *ler* is among the LEE genes that showed reduced expression in the *uvrY* mutant strain, we further determined whether UvrY could affect the activity of the *ler* promoter. As shown in Figure 4A, LEE1-*lux* activity was remarkably reduced upon *uvrY* deletion, and the signal was recovered back to the WT level after *uvrY* was complemented (Figure 4A). These results indicated that UvrY influences the activity of the *ler* promoter.

To further investigate whether the effect of UvrY on *ler* promoter activity was direct or indirect, an electrophoretic mobility shift assay (EMSA) was performed with the *ler* promoter and purified UvrY-His_6_ (Figure 4B). A clear mobility shift band was observed for the *ler* promoter, along with an increase in the concentration of UvrY-His_6_ (Figure 4C), suggesting a direct interaction between UvrY and P_LEE1_. To verify this, a positive control gene *csrC*, which has been well-characterized as the target of UvrY in *E. coli* and *Salmonella enterica* strains [36], was also selected for EMSA detection, and similar results were obtained (Figure 4D). However, no retarded bands were detected in the promoter regions of LEE2/3, LEE4, and LEE5 (Appendix A), or the negative control *rpoS* coding region (Figure 4E). These results indicated that UvrY could directly bind to the *ler* promoter in vitro. To further understand the UvrY binding activity at the *ler* promoter in vivo, a chromatin immunoprecipitation (ChIP)-qPCR assay was performed. As shown in Figure 4F, the UvrY proteins were enriched at both the *ler* and *csrC* promoters (positive control), as they exhibited a 2.1- and 4.5-fold higher signal in the ChIP samples than in the mock-ChIP samples (Figure 4F). Conversely, the fold enrichment of the promoter regions of LEE2/3, LEE4, and LEE5, or the *rpoS* coding region was not significantly different between the ChIP and mock-ChIP samples (Figure 4F). Taken together, these results demonstrated that UvrY can activate *ler* gene expression by directly binding to its promoter, thereby positively regulating *ler* and LEE gene expression in EHEC O157:H7 cells.

### 2.5. UvrY Affects Other Virulence Gene Expression in EHEC O157:H7

Given the above findings that the regulator UvrY promotes EHEC O157:H7 adherence and colonization through the transcriptional activation of LEE genes via direct activation control at *ler*, we wondered whether UvrY is implicated in other biological processes through transcriptional modulation. To achieve this, a high-throughput RNA-seq experiment and gene expression analysis were performed on the WT and Δ*uvrY* strains cultured in the same virulence-inducing DMEM (Figure 2). Significantly, a total of 559 genes showed differential expression when considering *p*-values ≤ 0.05 and a |log_2_foldchange| ≥ 1 between the WT and Δ*uvrY* (Figure 5A). Overall, 314 downregulated genes and 245 upregulated genes were identified in the Δ*uvrY* strain (Figure 5A, Appendix A). Among these differential expression genes (DEGs), five downregulated and five upregulated genes were randomly selected and validated using qRT-PCR, which demonstrated consistent gene expression changes (Figure 5B), indicating the repeatability and good quality of the RNA-seq data for further analysis. Notably, almost all the LEE pathogenic island genes (40 of 41 genes) were downregulated (2.1–11.7 fold) in the RNA-seq profile (Figure 5C), pinpointing our characterized UvrY functions in EHEC pathogenicity promotion at previous sections (Figure 1, Figure 2, Figure 3 and Figure 4).

In addition to the *ler* and LEE genes, several other virulence genes were detected with gene expression changes upon *uvrY* deletion, including the downregulated *nleA* and *ehxCABD* operon genes, as well as the upregulated genes *hdeA* and *hdeB* (Figure 5C, Appendix A). NleA is a non-LEE-encoded effector in EHEC that plays a role in the suppression of host inflammasome activity by targeting NLRP3 [37,38]. The *ehxCABD* operon gene *exhA* is responsible for the production of enterohemolysin, which has been identified as a major virulence factor in various bacterial pathogens [39]. These results provide clues suggesting that more flexible downstream targets are regulated by UvrY. In contrast, *hdeA* and *hdeB* were upregulated in the Δ*uvrY* strains (Figure 5C). HdeA and HdeB are important for *E. coli* survival in acidic environments [40]. As the colonic environment is neutral or slightly alkaline [41], it is speculated that the reduction in *hdeA*/*hdeB* expression is an energy-saving strategy for EHEC to adapt to the colonic niche, while *uvrY* depletion may disrupt such regulation and, hence, result in decreased survival capability in the host colon. Collectively, the results suggested that in addition to our afore-identified Ler-dependent regulation role of UvrY in virulence, other UvrY-mediated virulence modulation pathways need to be confirmed and validated in the future.

### 2.6. UvrY Is Involved in the Regulation of Multiple Biological Processes in EHEC O157:H7

To obtain a global overview of the function of UvrY in gene expression changes, gene ontology (GO) analysis was performed for the 559 DEGs. Downregulated genes were enriched in translation, gene expression, peptide and cation transmembrane transport, drug metabolism, pathogenesis, and secretion biological pathways (−log_10_P_value_ ≥ 1). Vitamin and lipid biosynthesis pathways were enriched in the upregulated gene sets (Appendix A), whereas the carbohydrate catabolic process, lipid metabolism, and ATP metabolism pathways were distributed at both the up- and downregulated gene sets (Appendix A). These results suggested that, in addition to pathogenesis, UvrY has a global regulatory role involving diverse biological processes. A KEGG analysis was further performed to investigate the metabolic processes modulated by UvrY. Among them, the expression of biotin and butanoate metabolism genes was downregulated, whereas that of thiamine and folate metabolism genes were upregulated (Figure 5D and Appendix A), suggesting the possible involvement of TCS BarA/UvrY in signal sensing or environmental condition adaptation.

Overall, our work showed that the regulator UvrY not only activates transcription in LEE-encoded T3SS through a direct activation activity in *ler* but also modulates the expression of other non-LEE virulence-related genes. As a result, the EHEC O157:H7 strain without UvrY showed a decrease in cellular adherence, mouse colonization level, and capability. Therefore, our study demonstrated the novel function of TCS UvrY in promoting EHEC O157:H7 colonization in the host. Meanwhile, this study not only broadens our knowledge of EHEC virulence regulation but also provides potential novel targets for intervention and treatment strategies in EHEC infection inhibition.

## 3. Discussion

EHEC O157:H7 is a critical A/E pathogen responsible for outbreaks of bloody diarrhea, hemorrhagic colitis, and hemolytic uremic syndrome (HUS) [42,43]. In EHEC, the LEE pathogenicity island regulation is complex and orchestrated, which has formed an elaborate regulatory network for LEE gene expression centered on the Ler protein and involving different types of regulators [7,44]. In this study, we found that the regulator UvrY of the TCS BarA/UvrY is critical for EHEC pathogenicity. UvrY enabled EHEC O157:H7 to increase T3SS gene expression and facilitated EHEC O157:H7 colonization in the mouse colon in a Ler-dependent manner. Diverse biological processes have been characterized for TCS BarA/UvrY, including central carbon metabolism, flagellum biosynthesis, motility [25], LPS production, fimbriae formation, and other virulence-related genes [26,28,29]. Therefore, our results suggested a novel role for UvrY in the EHEC O157:H7 virulence regulatory pathway. However, EHEC O157:H7 is known to be evolutionarily closer to the enteropathogenic *E. coli* (EPEC) strain O55:H7 than to other non-O157 EHEC strains [45,46]. Whether the identified regulation mechanism of UvrY in EHEC O157:H7 is conserved in other serotypes of EHEC strains or EPEC O55:H7 needs to be explored later.

When EHEC O157:H7 reaches the colon, it precisely regulates the expression of pathogenesis-related genes via TCSs, sensing a broad range of environmental changes and causing severe symptoms [19,47]. As previously described, PhoP/PhoQ senses a low magnesium-induced signal to increase the EHEC O157:H7 LEE gene expression [20]; FusK/FusR senses the fucose and affects the expression of virulence genes [8], while QseB/QseC responds to the host epinephrine/norepinephrine signal to modulate the transcription of the flagella and motility genes, which are beneficial for the EHEC to travel in the intestinal epithelium and initiate the infection process [47]. Our RNA-seq data showed that the metabolism of several cellular components, including multiple vitamins, were affected after *uvrY* deletion; however, whether BarA/UvrY, as a member of TCSs, responds to these signals from the intestine to activate the transcription of virulent genes needs further exploration. Furthermore, the gene expression of the acidic adaptation factors HdeA and HdeB [40] was also modulated by UvrY, providing pH adaptation clues for EHEC O157:H7 living in the host colon.

In addition, a previous study reported that UvrY could inhibit RNA-binding protein CsrA activity [25,48]; CsrA has a negative effect on LEE gene transcription in EHEC O157:H7 [49]. Therefore, the multiple mechanisms of the UvrY function in the promotion of LEE gene expression were speculative. In addition to the direct binding of UvrY to the promoter region of *ler* to modulate LEE genes, whether UvrY could activate LEE genes indirectly by influencing the CsrA activity should be further confirmed. Meanwhile, numerous genes associated with central carbon and fatty acid metabolism were differentially expressed in both the Δ*uvrY* (this study, Appendix A) and Δ*csrA* RNA-seq data [49]. How these pathways are regulated by UvrY and whether a synergistic or hierarchical regulation mechanism exists between CsrA and UvrY warrants further analysis.

## 4. Materials and Methods

### 4.1. Strains and Plasmids

Bacterial strains and plasmids used in this study are listed in Appendix A. Oligonucleotides used in this study are listed in Appendix A. The mutant strains were generated using the λ-Red recombinase system, as previously described [50]. The *uvrY* complementary strain was obtained through cloning the *uvrY* gene on a pTRC99a plasmid, and the pTRC99a-*uvrY* plasmid was electroporated back into the Δ*uvrY* strains. The strains were cultured in LB broth with different antibiotics accordingly, and the utilized antibiotic concentrations were 100 μg/mL for ampicillin, 25 μg/mL for chloramphenicol, 50 μg/mL for nalidixic acid, and 50 μg/mL for kanamycin, respectively.

### 4.2. RNA Isolation, Purification, and Library Preparation for Sequencing 

For in vitro RNA isolation, EHEC O157:H7 WT and the Δ*uvrY* mutant strains were cultured in LB broth overnight at 37 °C, 180 rpm, and transferred to 20 mL of fresh DMEM (Thermo Fisher, Waltham, MA, USA) with an inoculation ratio of 1:100. When the culture reached the exponential growth phase (OD_600_ = 0.6–0.8), cells were collected by centrifugation at 5000 rpm for 5 min at 4 °C. For in vivo RNA isolation, the infected colon was collected and treated with liquid nitrogen grinding. Total RNA extraction was performed using the TRIzol reagent (Invitrogen, Waltham, MA, USA) according to the manufacturer’s instructions, and RNA samples were purified using the RNeasy mini kit (Qiagen, Germantown, MD, USA). RNA quality was determined by running a 1.2% denaturing agarose gel and further detected by the NanoDrop 2000 spectrophotometer (Thermo Fisher, Waltham, MA, USA) with RNA integrity. RNA library preparation and high throughput sequencing were performed by company Novogene, Inc. (Tianjin, China). The sequencing raw reads were mapped to EHEC O157:H7 str. EDL933 genome reference (genome version: ASM73296v1) using hisat2 (version: 2.1.0). All the sequencing data have been deposited in the NCBI SRA database under the accession codes SRR19262143-SRR19262148.

### 4.3. Quantitative RT-PCR (qRT-PCR)

The complementary DNA (cDNA) was generated from one μg of total RNA using the Primescript 1st strand cDNA synthesis kit (Takara, Shiga, Japan). The cDNA sample was diluted 3-fold prior to performing downstream experiments. qRT-PCR was performed using the Applied Biosystems 7500 real-time PCR system and SYBR green PCR master mix (Applied Biosystems, Waltham, MA, USA). All the data were normalized to levels of housekeeping gene *rrsH* [51]. The relative expression level of each gene was calculated using the cycle threshold method (2^−ΔΔCt^) [52]. At least three biological replicates were carried out for each experiment. All the oligonucleotides used for qRT-PCR are listed in Appendix A.

### 4.4. Growth Curve Detection

Strains were cultured overnight in LB broth at 37 °C, 180 rpm, and washed three times with phosphate buffered saline (PBS), and a 1:1000 dilution to the DMEM or LB broth at 96-well plates was further conducted. The measurement was performed in a multifunctional microplate tester (TECAN Spark, Shanghai, China), and parameters were set as follows: 37 °C, setting shake “On” and measured at OD_600_ for 24 h. Three independent experiments were conducted and analyzed.

### 4.5. Cultivation of HeLa Cells and Performing Bacterial Adherence Assay

HeLa cells were purchased from the Shanghai Institute of Biochemistry and Cell Biology of the Chinese Academy of Sciences (Shanghai, China) and cultured in DMEM with 10% fetal bovine serum at 37 °C under 5% CO_2_. The bacterial adherence assay was performed as previously described with some modifications [53]. Briefly, overnight bacterial cultures were 1:100 transferred in DMEM at 37 °C until the culture reached OD_600_ of 0.6–0.8. Cultures were washed three times with PBS and resuspended in fresh DMEM without fetal bovine serum. Bacterial cultures were added to culture dishes with a monolayer of HeLa cells at a multiplicity of infection (MOI) of 100, and 3 wells were set up for each strain in parallel for labeling. After 3 h, non-adherent bacteria were washed away with prewarmed PBS six times, and the remaining cells were subjected to lysis using 0.1% Triton X-100 (Sigma-Aldrich, Shanghai, China). The lysed cell suspension was diluted with PBS in 10-fold serial dilutions, and the lysates were spread on LB agar plates for bacterial CFU counting.

### 4.6. Fluorescent Actin Staining (FAS)

Fluorescent actin staining assays were performed as previously described with slight modifications [54]. Briefly, before cell suspension was added to a 6-well plate, sterile tweezers were used to place sterile cell coverslips into a 6-well plate. Cell culture conditions and co-bacteria culture times were same as those described in the bacterial adhesion assay. After incubation for 3 h, the coverslips were washed with PBS six times, fixed with 4% paraformaldehyde, and permeabilized with 0.1% Triton-X-100. The cells were stained with fluorescein isothiocyanate-labeled phalloidin (Sigma-Aldrich, Shanghai, China) to visualize actin filaments and stained with propidium iodide (PI) (Beyotime, Shanghai, China) for nucleus visualization. At least 50 HeLa cells were calculated for A/E lesions forming number counting for each bacteria strain.

### 4.7. Western Blotting

Overnight bacterial cultures were 1:100 transferred in DMEM until the culture reached an OD_600_ of 1.0. The bacteria were collected and washed three times with PBS, and then the cells were lysed by ultrasound. Equal amounts of total proteins (40 μg) were loaded on a 12% SDS-PAGE gel and transferred onto a polyvinylidene difluoride (PVDF) membrane. The blot membrane was blocked with QuickBlocK^TM^ blocking buffer (Beyotime, Shanghai, China) for 30 min at room temperature, followed by incubation with polyclonal antisera (mouse) against Tir or intimin and with primary antibody (1:10,000 dilutions for anti-DnaK antibody; Abcam, Cambridge, UK) for 2 h and another 1 h of incubation with secondary antibody (1:5000 dilutions for the goat anti-mouse-HRP; CWBIO; Beijing, China). Blots were detected using the ECL-enhanced chemiluminescence reagent (CWBio, Beijing, China), and protein levels were quantified using ImageJ software (NIH, Version: 1.8.0). Three independent biological replicates were performed for each experiment.

### 4.8. Luminescence Screening Assay

Luminescence screening assay was performed as previously described with some modifications [55]. The *ler* promoter (P_LEE1_) was cloned into the pMS402 plasmid using Gibson assembly (NEB Cat. E2621S). The fusion reporter plasmid P_LEE1_-pMS402 was transformed into the relative bacteria and cultured in DMEM until OD_600_ reached 0.6–0.8. The promoter activity of P_LEE1_ (LEE1*-lux)* was measured as counts per second (cps) of light production. The *lux* activity value was quantified by CPS/OD_600_.

### 4.9. Chromatin Immunoprecipitation and Quantitative PCR (ChIP-qPCR)

The ChIP assay was performed as previously described [56]. The *uvrY*-3 × Flag-pTrc99a recombinant plasmid was electroporated into Δ*uvrY* strains and cultured in LB broth at 37 °C, 180 rpm until OD_600_ reached 0.6–0.8. Cross-linking was performed with 1% formaldehyde for 25 min at room temperature, followed by 0.5 M glycine incubating for 5 min to quench the cross-linking. The bacteria were harvested by centrifugation, and the pellets were washed three times in PBS. Pellets were resuspended in 500 μL lysis buffer (Tris-HCl 50 mM; pH 7.5, NaCl 100 mM, EDTA 1 mM, lysozyme 20 mg/mL, RNase A 0.5 mg/mL, PMSF 1 mM) at 37 °C for 30 min, and 500 μL sonication buffer (Tris-HCl 100 mM; pH 7.5, NaCl 200 mM, EDTA 1 mM, TritonX-100 2%) was further added. The lysate was sonicated with 16 cycles of 25 s on/off at 95% amplitude. Cell debris was removed using centrifugation at 12,000× *g* for 10 min. The generated DNA fragments’ size in supernatant for downstream IP experiment was approximately 300–500 bp. The supernatant was aliquoted into two tubes (500 μL for each), one of which was added with 20 μL anti-FLAG antibody (Sigma-Aldrich, Shanghai, China) and labeled as ChIP sample, and the other without addition of any antibodies was labeled as mock-ChIP sample. Both the mock-ChIP and ChIP samples were incubated with the 50 μL protein A magnetic bead (MCE, Plainsboro Township, NJ, USA) and incubated in the rotary strapping machine at 4 °C for 5 h. The magnetic beads were washed three times with sterile PBS buffer and re-suspended in 300 μL elution buffer (Tris-HCl 50 mM; pH 8.0, EDTA 10 mM, SDS 1%). The eluted samples were de-cross linked at 65 °C for 3 h, followed by adding 10 μL RNase A (10 mg/mL) for RNA decontamination. The eluted DNA samples were further purified using DNA purification kit (Qiagen, Germantown, MD, USA). To investigate the enrichment folds of the target gene fragment in the ChIP sample relative to mock-ChIP sample, qRT-PCR was performed, and the experiment procedure and data analysis were conducted as described in previous section. The experiments were independently performed at least three times for data analysis.

### 4.10. Electrophoretic Mobility Shift Assay (EMSA)

The inducible-type pET28a-*uvrY* plasmid was transformed into *E. coli* BL21 (DE3) strain, and the UvrY-His_6_ fusion protein was expressed and purified from the lysate supernatant of the BL21 using a Ni^2+^-NTA His-Bind resin (QiHai, Shanghai, China). DNA fragments of the LEE1, LEE2/3, LEE4, LEE5, *csrC* promoter regions, and *rpoS* coding region were PCR-amplified and purified using a gel DNA extraction kit (Sparkjade; Jinan, China). The UvrY-His_6_ protein at concentrations ranging from 0 to 2 µM were incubated with 30 ng of above DNA fragments (P_LEE1_, P_LEE2/3_, P_LEE4_, P_LEE5_, P*_csrC_*_,_
*a*nd *rpoS* coding region) at 25 °C for 30 min, separately, in a 20 μL reaction tube containing binding buffer (20 mM Tris HCl; pH 7.5, 50 mM KCl, 3 mM MgCl_2_, 0.1 mM dithiothreitol, 10% glycerol, and 20 mM acetyl phosphate). The protein-DNA reaction samples were then loaded onto a 6% polyacrylamide gel immersed in 0.5 × Tris-Borate-EDTA (TBE) buffer for electrophoresis. DNA fragments were stained with Gel Red (Beyotime, Shanghai, China), and the photos were captured in Gel Images system (Tanon, Shanghai, China).

### 4.11. Mice Colonization Assay

Six-week-old female BALB/c mice were used for conducting the intrarectal infection experiment. Mice were subjected to fasting for 22 h prior to infection. The bacterial cells were collected by centrifugation and suspended in PBS. In each group, mice were orally infected with 10^9^ CFU of bacteria in 100 μL PBS. The infected mice were anaesthetized and euthanized via cervical dislocation at 6 h after infection. The distal specimens of the colon were harvested, and the colon contents were squeezed out and then weighed. For bacterial CFU counting, each colon specimen was ground and homogenized using PBS. The homogenates were diluted, and bacterial suspensions were spread on LB agar plates containing the corresponding antibiotics for bacterial CFU counting.

### 4.12. Statistical Analysis

Data were analyzed using GraphPad Prism (version 7.00; La Jolla, CA, USA). The data presented in each figure or table represent mean values with standard deviation (SD) obtained from three independent experiments. Student’s *t*-test was applied for *p*-value calculation between each two experimental groups. The Mann–Whitney rank-sum test was performed to assess statistical significance in mouse experiments. We applied *Fisher’s* exact test and FDR multiple test correction to find the significant GO and KEGG categories. Differences were considered significant at *p* ≤ 0.05.

## Figures and Tables

**Figure 1 ijms-24-02297-f001:**
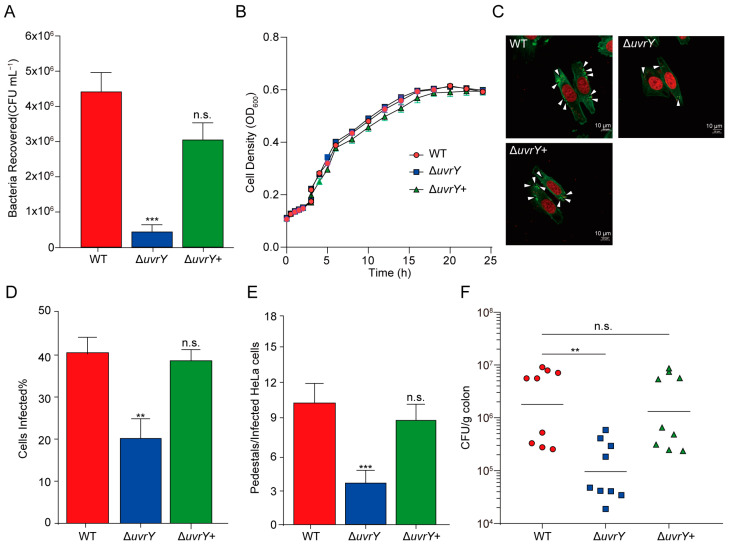
UvrY positively affects the adherence capability of EHEC O157:H7 to the host. (**A**) A bar plot showing the adherence ability of WT, Δ*uvrY*, and Δ*uvrY*+ strains to HeLa cells (*n* = 3). (**B**) A line plot showing the growth curves of WT, Δ*uvrY*, and Δ*uvrY*+ strains in DMEM for 24 h (*n* = 3). (**C**) Photos showing the HeLa cells infected with WT (left), Δ*uvrY* (right), and Δ*uvrY*+ (below) strains after 3 h post-infection. Green color represents the staining with fluorescein isothiocyanate-labeled phalloidin for actin visualization; red color represents the staining with proidium iodide for nuclei visualization. A/E lesions were indicated by white arrowheads in each photo. Scale bar is 10 μm. (**D**) A bar plot showing the quantification of infected HeLa cells percentage after co-culture with WT, Δ*uvrY*, and Δ*uvrY*+ strains (*n* = 50). (**E**) A bar plot showing the pedestals formation numbers in each infected HeLa cell by WT, Δ*uvrY*, and Δ*uvrY*+ strains. Fifty cells were counted for each strain (*n* = 50). Significance was determined using two-tailed unpaired Student’s *t*-test (**A**,**D**,**E**). (**F**) A scatter plot showing the mice colonization bacteria CFU numbers after inoculation with WT, Δ*uvrY*, and Δ*uvrY*+ strains at 6 h post-infection (*n* = 9). Horizontal bars represent the median. Significance was determined by Mann–Whitney rank-sum test. ** represents *p*  ≤  0.01; *** represents *p*  ≤  0.001; n.s. represents no significant difference.

**Figure 2 ijms-24-02297-f002:**
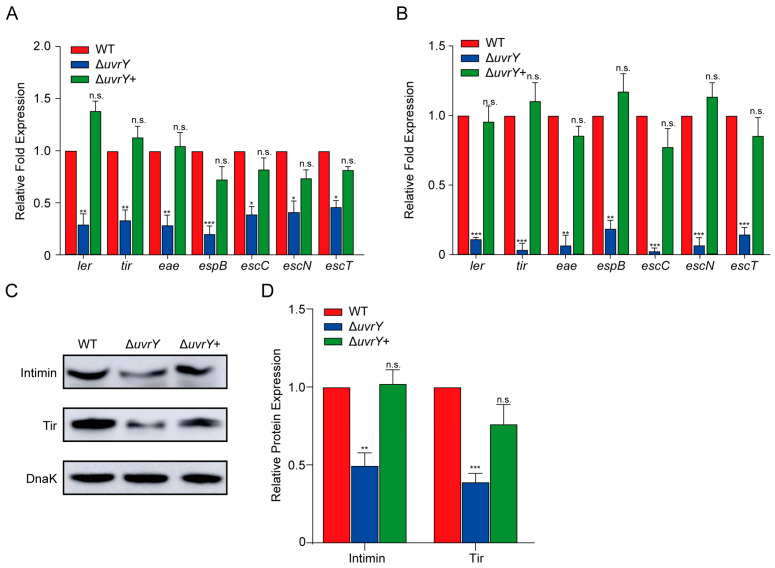
UvrY activates LEE gene expression in EHEC O157:H7. (**A**,**B**) Bar plots showing the relative expression level of seven representative LEE genes measured using qRT-PCR in WT, Δ*uvrY*, and Δ*uvrY*+ strains (**A**) cultured in DMEM for 3 h (*n* = 3) or (**B**) isolated from mice post-infection for 6 h. (**C**) Western blot analysis of the expression of intimin and the translocated intimin receptor (Tir) in WT, Δ*uvrY*, and Δ*uvrY*+ strains cultured in DMEM. DnaK was loaded as a quantification control. (**D**) A bar plot showing the quantification level of proteins in (**C**), performed in ImageJ. Three independent experiments were performed for Western blot analysis. Significance was determined using two-tailed unpaired Student’s *t*-test (**A**,**B**,**D**). * represents *p*  ≤  0.05; ** represents *p*  ≤  0.01; *** represents *p*  ≤  0.001; n.s. represents no significant difference.

**Figure 3 ijms-24-02297-f003:**
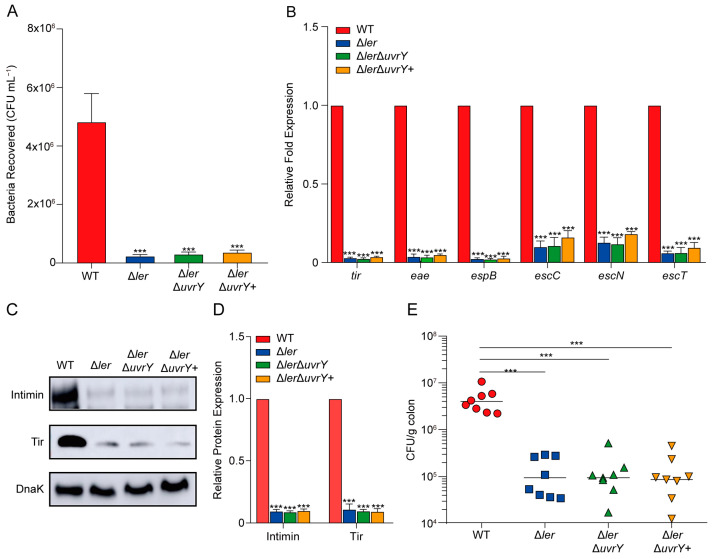
UvrY activates LEE gene expression via *ler*. (**A**) A bar plot showing the adherence level of WT, Δ*ler*, Δ*ler*Δ*uvrY*, and Δ*ler*Δ*uvrY*+ strains to HeLa cells *(n* = 3). (**B**) A bar plot showing the qRT-PCR expression quantification level of the representative LEE genes in WT, Δ*ler*, Δ*ler*Δ*uvrY*, and Δ*ler*Δ*uvrY*+ strains (*n* = 3). (**C**) Western blot analysis of intimin and Tir in WT, Δ*ler*, Δ*ler*Δ*uvrY*, and Δ*ler*Δ*uvrY*+ strains cultured in DMEM for 3 h. DnaK was loaded as a quantification control. (**D**) A bar plot showing the quantification level of proteins in (**C**), performed in ImageJ. Three independent experiments were performed for the Western blot analysis. Significance was determined using two-tailed unpaired Student’s *t*-test (**A**,**B**,**D**). (**E**) A scatter plot showing the mice colonization bacteria CFU numbers after inoculation with WT, Δ*ler*, Δ*ler*Δ*uvrY*, and Δ*ler*Δ*uvrY*+ at 6 h post-infection (*n* = 8). For each group, eight BALB/c mice were utilized. Significance was determined using Mann–Whitney rank-sum test. *** represents *p*  ≤  0.001.

**Figure 4 ijms-24-02297-f004:**
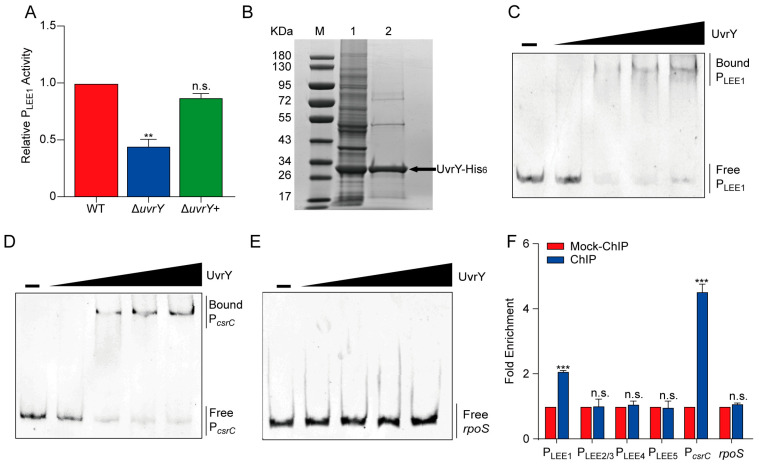
UvrY directly binds to the promoter of the *ler*. (**A**) A bar plot showing the LEE1-*lux* activity in WT, Δ*uvrY*, and Δ*uvrY*+ strains, respectively. LEE1-*lux* activity values in WT were normalized to 1. (**B**) SDS-PAGE photograph of the purified UvrY-His_6_. Lane M: protein marker; Lane 1: total protein; Lane 2: purified UvrY-His_6_ protein. (**C**–**E**) Chemiluminescence photographs showing the EMSA result for the binding affinity of protein UvrY-His_6_ with (**C**) the *ler* (LEE1) promoter; P_LEE1_, (**D**) *csrC* promoter; P*_csrC_*, and (**E**) *rpoS* coding region. (**F**) A bar plot showing the relative fold enrichment of the promoter fragments of LEE1, LEE2/3, LEE4, and LEE5 in ChIP samples compared with mock-ChIP samples, measured using q-PCR. P*_csrC_* and *rpoS* were calculated as a positive or negative control, respectively. Significance was determined using two-tailed unpaired Student’s *t*-test (**A**,**F**). ** represents *p*  ≤  0.01; *** represents *p*  ≤  0.001; n.s. represents no significant difference.

**Figure 5 ijms-24-02297-f005:**
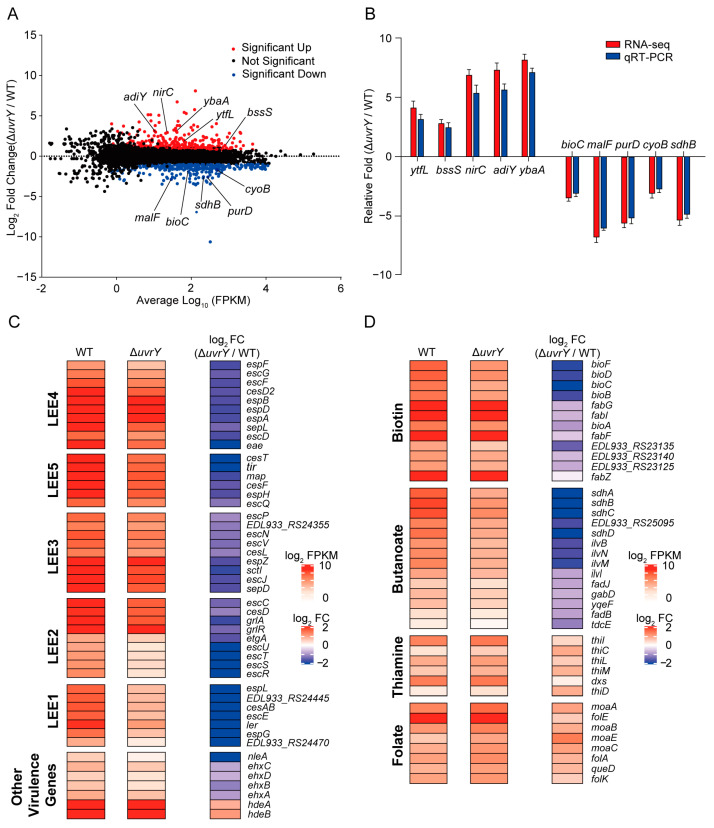
Transcriptome analysis of *uvrY* deletion measured using RNA-seq. (**A**) A MA plot showing all the differential expression genes between the WT and Δ*uvrY* strains cultured in DMEM. Each dot denotes one gene. Red dots represent upregulation genes (*p* ≤ 0.05, log_2_foldchange ≥ 1), and downregulated genes (*p* ≤ 0.05, log_2_foldchange ≤ −1) are marked with blue color. (**B**) A bar plot showing the qRT-PCR result of the ten randomly selected DEG genes in Δ*uvrY* compared with WT marked in (**A**), (*n* = 3). (**C**) A box plot showing LEE genes, *nleA*, *ehxCABD* operon, and *hdeA*/*hdeB* gene expression level in Δ*uvrY* compared with WT. (**D**) Box plots showing the expression changes in the biotin, butanoate, thiamine, and folate metabolism gene sets in Δ*uvrY* compared with the WT strain captured from the RNA-seq profile.

## Data Availability

The RNA sequencing data generated in this study are available in the NCBI SRA database under the accession codes SRR19262143-SRR19262148 (https://www.ncbi.nlm.nih.gov/sra/PRJNA839229, accessed on 18 May 2022). Other data are presented within the manuscript and Appendix A.

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
