# Peer review of "A Novel Role of the Two-Component System Response Regulator UvrY in Enterohemorrhagic *Escherichia coli* O157:H7 Pathogenicity Regulation"

_ijms, 2023, doi:10.3390/ijms24032297_

Round 1
Reviewer 1 Report
I have read the article entitled “A novel characterized role of the two-component system response regulator UvrY in enterohemorrhagic Escherichia coli O157:H7 pathogenicity regulation” by Wu et al. The manuscript describes the role of UvrY, the regulator protein of the BarA/UvrY system, in EHEC O157:H7 adherence and colonization capacity. In addition, it was demonstrated that UvrY increase expression of LEE genes by binding to ler promoter and transcriptomic analyses showed significant changes in transcript levels of a wide repertoire of genes when urvY is knocked. I think the manuscript is a solid piece of work.
Comments:
1- I suggest to remove the word “characterized” from the title.
2- Line 103. “…the growth curves were measured before and after uvrY deletion” Please rewrite this sentence. I think I got the point, but I guess the experiment was performed at the same time (in parallel) for wild type and mutant strains.
3- Line 220. Change for “…reduced expression in the uvrY mutant strain”
4- Line 221. If I understood the message, the sentence would be “…could affect the activity of the ler promoter” (same at line 224)
5- Figure 4. Please include an SDS-PAGE photograph of the purified UvrY-Hisx6. This would strength the results.
